# Tumor Cell Invasion in Glioblastoma

**DOI:** 10.3390/ijms21061932

**Published:** 2020-03-12

**Authors:** Arabel Vollmann-Zwerenz, Verena Leidgens, Giancarlo Feliciello, Christoph A. Klein, Peter Hau

**Affiliations:** 1Department of Neurology and Wilhelm Sander-NeuroOncology Unit, University Hospital Regensburg, 93053 Regensburg, Germany; arabel.vollmann@ukr.de (A.V.-Z.); VLeidgens@novocure.com (V.L.); 2Fraunhofer-Institute for Toxicology and Experimental Medicine, Division of Personalized Tumor Therapy, 93053 Regensburg, Germany; giancarlo.feliciello@item.fraunhofer.de (G.F.); christoph.klein@ukr.de (C.A.K.); 3Experimental Medicine and Therapy Research, University of Regensburg, 93053 Regensburg, Germany

**Keywords:** glioma, glioblastoma, migration, invasion, microenvironment, genetic program

## Abstract

Glioblastoma (GBM) is a particularly devastating tumor with a median survival of about 16 months. Recent research has revealed novel insights into the outstanding heterogeneity of this type of brain cancer. However, all GBM subtypes share the hallmark feature of aggressive invasion into the surrounding tissue. Invasive glioblastoma cells escape surgery and focal therapies and thus represent a major obstacle for curative therapy. This review aims to provide a comprehensive understanding of glioma invasion mechanisms with respect to tumor-cell-intrinsic properties as well as cues provided by the microenvironment. We discuss genetic programs that may influence the dissemination and plasticity of GBM cells as well as their different invasion patterns. We also review how tumor cells shape their microenvironment and how, vice versa, components of the extracellular matrix and factors from non-neoplastic cells influence tumor cell motility. We further discuss different research platforms for modeling invasion. Finally, we highlight the importance of accounting for the complex interplay between tumor cell invasion and treatment resistance in glioblastoma when considering new therapeutic approaches.

## 1. Introduction

Glioblastoma (GBM) is the most prevalent and malignant primary brain tumor of adults, with a mean incidence of 7 cases per 100,000 per year and a mean overall survival of about 16 months [1]. GBM is a biologically heterogenous tumor that exhibits all of the classic hallmarks of cancer [2], with significant differences between patients. Radio-chemotherapy, supplemented with tumor treating fields (TTFields), is the standard of care [3,4] leading to a mean overall survival of 20.9 month [4]. Personalized therapies that target specific pathogenic events or molecular targets have not been discovered, though subtypes of GBM being prognostically relevant have been defined by genetic [5] and epigenetic [6] methods. 

One of the clinical hallmarks of GBM is extensive infiltration of the tumor surrounding parenchyma [7,8]. Gliomas almost never metastasize out of the brain. Invasion happens along pre-existing structures such as blood vessels, white matter tracts and the subarachnoid space [9] and is likely coordinated by specialized cells that may lead collective invasion [10,11]. GBM cells can cross tissue barriers by remodeling their own cytoskeleton and the extracellular matrix (ECM) [9] and invade as individual cells [7] or collectively [12,13]. 

Reinvasion of distant GBM cells to the primary tumor location and invasion to distant areas may lead to colonization of these sites, thus triggering local and distant bulky relapse. Recurring tumors share many early tumor mutations with their progeny [14], indicating that phylogenetically primary tumor progenitor clones reside in niches from which they escape their dormant state to repopulate locally [15]. 

Several patterns of invasion have been described [16,17] which depend on (often conserved) genetic programs [18] and interactions with the microenvironment [10]. Invasion can be simulated with GBM brain tumor initiating cells (BTIC) [19,20] as well as differentiated cells [21,22]. 

Though the mechanisms leading to GBM invasion are increasingly understood in vitro and in animal models, results from the laboratory have not been translated into clinically relevant therapeutic approaches so far [23]. Within this review, we will provide an overview of the current knowledge on GBM cell invasion and provide guidance for future research in this field (Figure 1).

## 2. Conserved Genetic Programs to Coordinate Invasion in Glioblastoma

GBM BTIC and differentiated cells may exploit conserved genetic programs that are fundamental during neuro- and gliogenesis to migrate through the brain parenchyma [24]. These programs may determine the morphology of invading GBM cells and thereby their ability to move through tight spaces and to react to signals from other prevalent cells and extracellular fluids that are found within the microenvironment.

Migration of both neural progenitor and glioma cells is guided by specific structures, such as white matter tracts [25] and blood vessels [26,27]. Both types of cells typically exhibit a uni- or bipolar morphology that resembles the growth cones of outgrowing axons [28]. Several genes common in normal as well as in GBM progenitor cells have been described that shape cells to make them fit for invasion. One example is myosin II that plays a role in soma translocation of neural progenitor cells during brain development [29]. In concert with dynein, myosin II remodels the microtubule network in a fashion that locates centromeres and the nucleus in direction of the leading process [29]. Similarly, myosin II is also found in the leading, microtubule-rich protrusions of glioma cells, which facilitate invasion and cell-to-cell communication. [30]. Neurodevelopmental molecules necessary for neurite outgrowth and axon guidance are frequently repurposed by glioma cells. Chemo-attractants as well as -repellents show similar effects on normal as well as tumor cells. Pleiotrophin (PTN), for example, is a chemoattractant which normally promotes neurite outgrowth and neuroblast migration. It is produced by neural precursor as well as glioma cells; in the latter, PTN expression levels are inversely correlated with patient overall survival [31]. Genetic knockdown of PTN, on the other hand, strongly reduces glioma invasion [32].

## 3. Heterogeneity and Plasticity Within Glioblastoma

Clinical evidence suggests that some GBMs are more disseminated than others [33]. GBMs show remarkable interpatient and intratumoral heterogeneities on the genomic [34] and gene expression levels [35,36,37], leading to distinct phenotypes that may likely explain distinct behaviors. For example, mesenchymal-subtype GBM show an increased potential to invade in comparison to its neural, proneural and classical counterparts [5].

This general heterogeneity is mirrored by BTIC that possess stem cell characteristics [38,39] and are considered the most relevant cell population to drive GBM pathogenesis [40]. We and others have shown that BTIC are characterized by self-renewal, clonogenicity and differentiation potential into several lineages [41], and are able to induce tumors in animal models [42,43]. The phenotype of BTIC is primarily coined by their epigenetic state, and further modified by a number of transcription factors, chromatin regulators and associated cellular networks [44]. 

A number of transcription factors were identified as key regulators of transdifferentiation from an epithelial-like to a mesenchymal state (epithelial–mesenchymal transition, EMT) in a variety of different tumor types, including glioma. [36,45,46]. During EMT, tumor cells upregulate mesenchymal markers [47], lose their epithelial shape, acquire a spindle-like phenotype and thus acquire the ability to invade [48]. At suitable niches, EMT-transformed cells may convert back to an epithelial type, enabling the formation of a new tumor bulk [7,48].

Even though, in glioma, transition to a mesenchymal state emerges rather from a proneural than from an epithelial phenotype, similar factors contribute to this plasticity. Among the most prominent TFs involved in glioma EMT are ZEB1 [49,50], TWIST1 [51], SNAI1/2 [52,53], TAZ/WWTR1 [54] and the so-called “master regulators“ C/EBPβ and STAT3. The latter were shown to induce a mesenchymal gene expression signature and reprogram neural progenitors to a mesenchymal phenotype [36]. Recently, S100A4, a calcium binding protein, has been proposed as an upstream regulator of C/EBPβ and SNAI2 and as a critical mediator of mesenchymal transition in GBMs [50]. In addition, NFκB activation by TNFα was shown to induce mesenchymal differentiation of proneural BTICs via STAT3 and C/EBPβ [55]. 

Recent publications clearly indicate that glioma subtypes are subject to high plasticity [55,56]. Upon recurrence, a transition to the mesenchymal subtype is often observed and associated with worse overall survival [56,57]. This shift can be rapidly induced by therapeutic drugs or irradiation [58,59,60], which implies that subtype-tailored treatments might lose their activity. 

To add even more complexity to the matter, it is not clear for many of the involved processes whether they are predominantly regulated at the genomic, transcriptional, proteomic or metabolomic level. Recently, single-cell profiling has become available for DNA [61], RNA [62], the epigenome [63], proteins [64] and even multiple levels in parallel [65], raising hope that the complexity of processes involved in GBM heterogeneity and plasticity will be soon understood in even more detail.

## 4. Bulk Vs. Single Cell Analysis

A full understanding of the highly complex GBM ecosystem that drives infiltration may be masked by bulk analysis of the whole tumor. High resolution-analyses at the single cell level may better depict relevant molecular events [66]. On the other side, such information has to be reintegrated to understand the systems-related coordination of tumor invasion. 

Reflecting the method of analysis used, it is not entirely clear at this time if single leading cells or the tumor bulk coordinate invasion; several distinct patterns of invasion have been described in GBM [16,17] which relate to genetic programs [18] and factors from the microenvironment [10]. Single cell invasion—that can be amoeboid or mesenchymal [67], in the latter case with a spindle-like cell shape—is a common type of invasion in GBM and depends on a vivid turnover of cell–matrix interactions [68]. In contrast, collective invasion, the predominant pattern in solid tumors, also occurs in GBM, and separates further into multicellular streaming and collective invasion [17]. Guiding structures and chemoattractants [69] as well as a high contractility of cells [70] and close adherence within the tumor bulk may favor collective invasion. 

A large number of publications have addressed either single-cell or bulk high-throughput analysis (Table 1). Based on these papers, single-cell methods may be well suited to describe -omics signatures that relate to certain patterns of invasion, whereas bulk analyses are more suited to detect specific driver genes [66,71]. 

## 5. Mechanisms of Invasion that Relate to GBM cells 

GBM cells migrate in a highly coordinated manner [102]. Tumor cells are primarily adherent to adjacent cells, like neurons, astrocytes and endothelial cells, and to the ECM. Adherence is modulated by a variety of different proteins, for example, integrins or cadherins [103]. Integrins are sensitive sensors to the microenvironment and allow tumor cells to modify their attachment behavior [104]. They recruit adaptor molecules and proteins that relay signals into cells, e.g., phosphorylation and dephosphorylation events via the focal adhesion kinase (FAK) [105]. The ECM is shaped by proteolytic factors derived from GBM cells, such as matrix metalloproteinases [106,107], ADAMS [108] and cathepsins [109]. Matrix metalloproteinases (e.g., MMP-2 and MMP-9), enable GBM cells to detach from the microenvironment and to alleviate tension within the ECM [9]. While migrating, tumor cells adapt the composition of their cell membranes [110] and contract, with myosin II being of particular importance if migration happens through tight brain parenchymal spaces [111,112]. 

GBM cells also attract cells such as microglia, astrocytes and endothelial cells that secrete proteases to enhance migration [113]. By moving along blood vessels, GBM tumor cells degrade astrocyte connections to the basement membrane and thereby change the homeostasis in proximity of the blood vessels [114]. The focus of attention has shifted increasingly to communication between GBM cells and other cells in their microenvironment after so-called tumor microtubes have been detected recently that allow precise communication in between tumor cells and their neighbors [30,115].

Invasive programs are coordinated by overexpression of transcription factors like STAT3 and C/EBPβ, which have been shown to be synergistic initiators and master regulators of mesenchymal transformation in BTICs [36]. NF-κB may enhance and c-Myc may decrease the invasive activity in the migratory cell population [94]. Accordingly, promoter analysis of genes upregulated in invasion-leading cells revealed binding sites for NF-κB and C/EBPβ [116]. Additional programs that may influence invasion have also been described [45,46].

## 6. Mechanisms of Invasion that Relate to the GBM Microenvironment 

GBM invasion results from a continuous bidirectional interaction between tumor cells and their microenvironment [117]. The microenvironment consists of several ECM components, fluids, soluble factors such as chemokines and cytokines, and cells such as neurons, astrocytes, oligodendrocytes, endothelial cells and immune cells [118].

### 6.1. Extracellular Matrix 

ECM molecules provide structural support and act as a guiding scaffold or barrier [21,119]. The brain parenchyma, in contrast to other ECMs, lacks rigid components such as collagens, fibrinogen and laminin, and instead consists of proteoglycans [120], hyaluronic acid [121] and tenascin-C [120]. 

We and others have shown that the ECM itself, as well as cell–cell and cell–ECM connecting molecules and proteases, have an impact on glioma cell migration [122,123,124,125]. Soluble factors such as growth factors [126] also guide invasion and contribute to the fate of BTICs and differentiated tumor cells [127]. Several chemoattractants such as bradykinin [128] and vascular endothelial growth factor (VEGF) [129] play relevant roles in this context. 

### 6.2. Adapted Niches within the Brain Parenchyma

The composition of the ECM generates specific stem cell niches and facilitates migration and invasion in distinct ways [130,131]. 

The most prominent example of a niche within the diseased brain is the hypoxic niche. A hypoxic environment stimulates the expression of the hypoxia-inducible factors (HIFs) HIF1α and HIF2α that favor an invasive phenotype [132]. An acidic environment also stimulates HIF function [133] and activates MMPs [134], therefore inducing similar mechanisms.

Another example of a brain-intrinsic niche that may enhance invasion is the angiogenic niche. Endothelial cells interact closely with BTICs and secrete factors that maintain these cells in a stem-cell-like state [127] and enable invasion [135]. 

### 6.3. Non-neoplastic Cells

Amongst the nonimmunogenic cell types that are prevalent in the brain, endothelial cells are important inducers of invasion, especially in the angiogenic niche. The activation of angiopoietin 1 and its crosstalk with Tie2 increases invasion [136].

The most important immunogenic cell type able to promote invasion in GBM are microglia. GBM progenitor cells closely interact with microglia [137] and thereby enable invasion, e.g., by induction of MMPs [138], induction of growth factors and release of ECM components [139]. Several proinvasive signaling pathways such as Pyk2 [140], osteopontin-cd44 [141], EGFR [142] and TGF-β [143] are also induced by the interaction of GBM with microglia. 

## 7. In Vitro Models to Investigate Invasion in Glioblastoma

In vivo animal models allow the study of GMB invasion within the microenvironment of a living host. However, they are often complex, expensive, time consuming, impaired by a low reproducibility and associated with difficulty in dissecting the different components involved in the process. 

Therefore, a number of in vitro models have been developed in recent years [144]. These models are constructed to mirror features of the human disease such as widespread dissemination of tumor cells and should cover sufficient timeframes to allow population of distant sites and repopulation of the primary tumor area [25]. In addition, they are autologous in the best case and feature not only BTICs and differentiated tumor cells, but also their microenvironment.

### 7.1. Two-Dimensional Models 

Two-dimensional models are based on monolayers of GBM tumor cells on surfaces that are coated by, for example, ECM molecules. These models are simple to use but have important limitations. Cells in monolayers do not invade (but migrate) and are separated from their natural microenvironment, therefore behaving differently [145]. Transwell assays allow a higher level of complexity but often produce a wide range of results with high standard deviations and have to be interpreted with caution [146]. In summary, no convincing two-dimensional model has been developed for live-imaging of invading human BTICs so far [147].

### 7.2. Three-Dimensional Models 

Three-dimensional models can be used to investigate intraparenchymal [114] and perivascular invasion [148]. Typically, collagen-based 3D assays are used, however these often lack essential brain ECM elements such as laminin, hyaluronic acid, proteoglycans and fibrous glycoproteins [149,150]. 

Tumor spheroids on the other hand are generated as organotypic multicellular spheroids that are obtained by culturing resected fragments of tumors [151]. They retain part of the original morphology [152] and reflect intratumoral heterogeneity [153]. Tumor organoids are another recently developed model to culture tumor cells within matrigel for up to several months. They also consist of heterogenous areas and mimic tumor development and gradients [154,155]. 

As a three-dimensional ex vivo model, organotypic slice cultures (OBSC) have been developed to allow long-term imaging of cancer cell migration and invasion [156,157,158,159] and to describe specific pathogenic events [160]. In addition, the OBSC-model allows the monitoring of tumor cell invasion in real time. OBSCs fill the gap between 2D cell cultures and in vivo studies and reduce the number of animals needed and the burden on them [159]. The model also allows the discrimination of events into distinct molecular and cellular levels [161] and to evaluate therapeutic interventions in situ. Functional cDNA libraries can be obtained from as few as a single cell extracted from OBSCs [11,162]. Immunohistochemical (IHC) stainings of OBSCs after as long as 21 days in situ allow to monitor relevant environmental factors [163]. 

The combination of an orthotopic GBM model and time-lapse in vivo microscopy of human brain tumor initiating cells through a cranial window allows in-depth analysis of tumor cell infiltration to acquire a better insight of GBM tumor cells invasion dynamics within their environment [20]. Mice can be imaged repeatedly [164], and images can be processed in a way that allows the description of features such as movement direction, persistence of direction, speed and velocity [20]. This model, specifically if allograft tumor cells are used, is close to nature, but is technically complex and puts a high burden on animals.

## 8. Emerging Novel Technologies and Approaches for Studying Invasive GBM

Models of tumor progression offer the possibility to predict the behavior of tumor invasiveness and, consequently, to define more accurate prognostic estimates and therapeutic strategies. In the last 20 years, several in silico models have been developed to elucidate the main mechanisms that induce glioma proliferation and invasion [165]. Although useful to investigate specific aspects of glioma invasion, many of these models have been derived from in vitro experiments, therefore limiting their use in clinical settings. 

The advent of high-throughput data generation, largely coincident with the development of next-generation sequencing, has boosted the knowledge of many cancers—including gliomas—allowing computational modeling of large datasets to assess and predict tumor dynamics. For example, the Cancer Genome Atlas (TGCA) and Repository for Molecular BRAin Neoplasia DaTa (REMBRANDT) are two large-scale efforts containing genomics, transcriptomics, DNA methylomics and proteomics data as well as clinical data [166,167]. Such high dimensional data have been used successfully to cluster gliomas in different groups and to identify known molecular characteristics commonly used in clinical evaluation [168]. 

Recently, the Ivy Glioblastoma Atlas Project (Ivy-GAP) has generated a transcriptional atlas for known anatomical features of GBM [169]. Starting from the Ivy-GAP spatial information and data derived from genome and transcriptome sequencing, it has been possible to generate computational models that are able to explain the dynamics of GBM [170]. It has thereby been shown that GBM follows a divergent evolution with substantial genomic differences between initial versus recurrent tumors, with the latter presenting more comprehensive and extensive alterations in, e.g., RTK-MAPK, RTK-PI3K, INK4-Rb and ARF-p53 pathways [14]. Similar, evolutionary gene expression patterns can lead to a more invasive phenotype, e.g., by activation of TGFβ by overexpression of LTBP4 [56]. These patterns can by even more dissociated by dissecting infiltrative GBM cells at a single cell level, rendering possible the identification of genes involved in the invasion of the interstitial matrix [87]. 

To understand infiltrating single cells on a deeper level, novel targeted approaches for single cell sequencing have been developed. For example, topographic single cell sequencing (TSCS) is a method that combines laser microdissection and cell catapulting [171] with single cell DNA sequencing, while preserving spatial information. The application of TSCS in mammary ductal carcinoma in situ (DCIS) has highlighted that initially rare invading clones shift the mutation frequencies of specific genes to better escape the basement membrane and establish an invasive tumor mass [172]. TSCS, if applied to GBM, could therefore be useful in the near future to more precisely study the gene signature of single infiltrating cells and to identify mechanisms of GBM invasion and resistance, as well as to pave the way for new drugs capable of targeting these cells. Information derived from these data could also suggest combinatorial therapeutic approaches by targeting multiple cell subtypes according to their molecular characteristics. 

Although DNA-seq and RNA-seq are useful to unravel cancer heterogeneity, clonal evolution and mechanisms of invasiveness in gliomas, they do not provide insight into cell-to-cell communications and interactions with the tumor microenvironment. Recently, spatial transcriptomics has therefore also interrogated cell-to-cell interactions in their natural context, revealing an unexplored landscape of heterogeneity in different types of cancers [173,174,175]. For example, spatial transcriptomics signatures in breast cancer have supported clinical diagnostics, offering the possibility to distinguish DCIS and invasive ductal carcinoma (IDC) [173], or to identify specific areas where certain transcripts (e.g., *TMSB10*) can predict an invasive phenotype [176]. An application of these techniques to GBM might therefore be of great relevance to explore tissue areas that surround the tumor core. Moreover, a comparison of spatial transcriptomics data over time in individual patients could give hints at which stage of the disease the invasive phenotype occurred, if known models of tumor evolution, e.g., branched or linear, are applied. 

Spatial transcriptomics may also unravel expression patterns in the inflammatory environment of cancers. Recently, a dissection of prostate cancer tissue has shown that reactive stroma and inflammation are present at early stages and promote tumor growth by activation of oxidative stress and integrin-linked kinase (ILK) signaling [174]. Similarly, gene expression gradients have been observed in malignant melanoma, where it has been shown that transcriptional programs in the cancer core overlap with surrounding lymphoid tissues [175]. Lastly, the application of spatial transcriptomics to synovial biopsies from patients with rheumatoid arthritis and spondyloarthritis has allowed to decipher inflammatory signatures and to predict immune cell patterns [177]. In GBM, heterogeneous populations of tumor-associated macrophages (TAM), regulatory T-cells and dendritic cells with immunosuppressive capacity have been recently described [178]. TAM thereby support tumor cell migration by producing a wide array of cytokines and growth factors in response to signals derived from glioma cells [179]. Nevertheless, it is still not clear which exact molecular mechanisms are at the basis of this interaction and if microglia and TAM acquire new phenotypes to support the invasion of glioma cells. The mentioned interplays could be better characterized if the cells would be analyzed in their spatial context.

A bioinformatic integration of the mentioned scRNA-Seq and spatial transcriptomics data has been recently developed and applied to infer the tissue architecture in nonglioma tumors, for example in pancreatic ductal adenocarcinoma [180]. Such integrated approaches have also allowed to reveal the cellular and spatial organization of bone marrow niches at an unprecedented resolution [181]. By integrating cell-type markers derived from scRNA-seq data, it is possible to determine relative cell-type proportions in spatial transcriptomics data and, possibly, to identify rare tumor cell types and patient-specific pathogenic factors. It is noteworthy that with improved resolution at single cell levels such approaches could also be applied in GBM to identify single-cell relationships in the tumor microenvironment to pave the way for more individualized therapy decisions. 

In the clinical setting, high-quality MRI data are nowadays transformed into radiomics, a method of use for building predictive models for diagnosis, prognosis and therapeutic response [182,183]. Radiogenomics—an advanced field of radiomics—aims to associate and overlay clinical and molecular features with imaging data in an attempt to describe intratumor heterogeneity [184,185]. As an example, overlapping of miRNA data derived from the TCGA with a MRI technique called fluid-attenuated inversion recovery (MRI-FLAIR) has shown concordance between peritumoral FLAIR intensity and downregulation of miRNA regulating genes involved in cellular migration and invasion, such as *POSTN* and *CXCL12* [186]. In addition, in patients with deep white matter track involvement and ependymal invasion, it has been observed that *MYC* overexpression and inhibition of NF-*K*B inhibitor-alpha (NFKBIA) support an invasive imaging signature [187]. Similarly, the correlation of imaging from nonenhancing parenchyma with the spatially matched genetic status has shown an association with copy number aberration (CNA) changes in driver genes of GBM, such as *PDGFRA*, *PTEN*, *TP53*, *RB_1_* and *EGFR* [188]. Further advances in radiogenomics could be of great help in the near future, especially in the follow-up phase of GBM when repeated biopsies or extensive genomic and transcriptomic analysis are not possible. In addition, 3D models that simulate response to radiotherapy have also been developed and validated in small number of patients, based on modern imaging techniques [189].

In summary, these multidimensional approaches may enable much better models and concepts for the clinical behavior of invasive GBM, allow to further fine-tune the landscape of an individual’s GBM and support the development of more precisely targeted and therefore personalized therapies.

## 9. Overcoming Treatment Resistance by Targeting Invasion in GBM

Molecular mechanisms of invasion and drug resistance overlap to a high extent and are influenced by cell-intrinsic and microenvironmental factors. An invasive tumor phenotype may therefore also increase resistance to antitumor systemic therapy and vice versa. Glioma cells stimulated to migrate showed downregulation of proliferation- and apoptosis-associated genes while genes involved in cell motility became overexpressed [190]. Similar results were shown by Molina et al., who found elevated MAPK signaling in the tumor core, corresponding to a proliferative state. Conversely, invading glioma cells downregulated MAPK and upregulated AKT, thereby activating prosurvival signaling cascades [191].

In general, members of pivotal signaling pathways such as p53, the Ras/MAPK, PI3K and mTOR pathways, growth factors, chemokines and integrins are hubs of mechanisms that jointly support tumor cell survival and resistance as well as migration and invasion. One prominent example is the tumor-suppressive transcription factor (TF) p53, which is altered in cancer, including glioma, via loss- or gain-of-function mutations. In both cases, apoptosis and cell cycle arrest are impaired via, e.g., downregulation of proapoptotic molecules like Bax and lack of p21 activation, respectively [192,193]. Concomitantly, epithelial-to-mesenchymal transition (EMT) is induced by upregulation of TFs Twist and Slug [194]. Another important example of such a pathway is the binding of integrins to ECM components that leads to downstream activation of GSK3, PI3K/AKT, STAT3 and FAK, resulting in enhanced survival [195,196] as well as chemo- and radio-resistance [196,197,198]. Cytoskeletal rearrangements are also induced via FAK, leading to altered ECM adhesion and motility [199]. There are many additional pathways and factors associated with therapy resistance and tumor cell motility, all of which highly relevant for both the induction of resistance and invasion. Most of them are summarized in recent excellent reviews [119,200].

In the future, it will be highly important to precisely define focal points that influence both invasion and resistance to therapy to select high priority candidates for a targeted drug intervention in patients with GBM. 

## 10. Conclusions

Malignant gliomas are, amongst other pathogenetic features, characterized by a unique ability to invade. From a clinical point of view, invasion renders gliomas microscopically nonresectable and presents a major obstacle for curative treatment. The immense interpatient and intratumoral heterogeneity of gliomas, as well as the pivotal role of the tumor microenvironment that makes a profound understanding of the involved mechanisms difficult, are increasingly appreciated in this context. Even so, research on the complex interplay between tumor cells of different subtypes and the surrounding brain parenchyma is still at its infancy. Current standard therapies are unable to target infiltrative tumor cells, and effective anti-invasive treatment strategies are not available so far. Therefore, there is urgent clinical need to investigate the mechanisms leading to invasion of glioma in more detail. Sophisticated sequencing approaches and advancing new in vitro and in vivo models will hopefully provide a better understanding of these interactions and soon present new targets for anti-invasive therapeutic strategies.

## Figures and Tables

**Figure 1 ijms-21-01932-f001:**
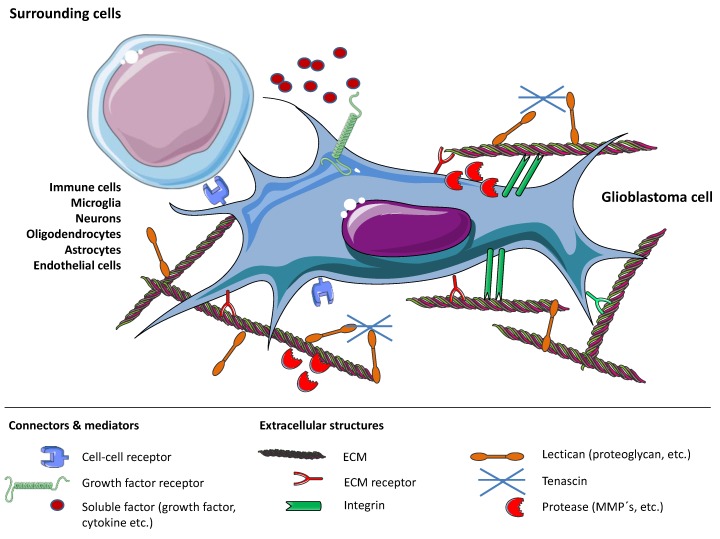
Overview of mechanisms that enable glioblastoma (GBM) cell invasion by shaping the tumor microenvironment. Processes involve the glioblastoma cell and extracellular matrix (ECM) themselves, as well as connecting molecules and proteases, soluble and bound factors and surrounding cells. Figure drawn with smart.servier.com.

**Table 1 ijms-21-01932-t001:** Publications with a focus on glioma cell invasion using bulk and single cell analysis. If not otherwise stated, human tumor material was used. Tumors were harvested from the tumor core, the tumor periphery, or without a specific allocation. Single cells or cells from the tumor bulk were harvested and used for further analysis. RNASeq, RNA-Sequencing; N.d., not described.

Author	Year	Model	Number of Tumors	Tumor Area	Single Cell Vs. Bulk	Material	Method	Identified Genes/Signatures Related to Invasion
[72]	2019	xenograft	n.d.	core *vs*. periphery	bulk	mRNA	RNASeq	TGF-β, THBS
[73]	2019	cell lines	n.d.	core *vs*. periphery	single-cell	mRNA	microarray	e.g., hypoxia, cell survival
[74]	2019	resection	17	core *vs*. periphery	bulk	mRNA	microarray	PDZ binding kinase
[75]	2019	resection, xenograft	n.d.	core *vs*. periphery	bulk	mRNA	RNASeq	CD109
[60]	2019	resection	28	n.d.	single-cell	mRNA	RNASeq	n.d.
[76]	2019	resection	144	n.d.	bulk	mRNA	RNASeq	Survivin, miR218
[77]	2019	cell lines	n.d.	n.d.	bulk	mRNA	microarray	CD164
[78]	2019	resection	n.d.	n.d.	bulk	mRNA	microarray	HOXB13
[79]	2019	cell lines	n.d.	n.d.	bulk	mRNA	microarray	UGP2
[80]	2018	resection	13	n.d.	bulk	mRNA	RNASeq	GLUD2
[81]	2018	resection	n.d.	n.d.	single-cell	mRNA	RNASeq	Wnt signaling
[82]	2018	cell lines	50	n.d.	bulk	mRNA	microarray	miR21, miR339, miR92b
[83]	2018	cell lines	n.d.	n.d.	bulk	mRNA	microarray	STAT3, slug
[84]	2018	resection	n.d.	n.d.	bulk	mRNA	microarray	SYK
[85]	2018	cell lines, resection	n.d.	n.d.	bulk	mRNA	microarray	FGF13
[86]	2018	resection	n.d.	n.d.	single-cell	mRNA	RNASeq	CSF1
[87]	2017	resection	4	core vs. periphery	single-cell	mRNA	RNASeq	e.g., cell survival, ATP generation, cell–cell adhesion
[88]	2016	resection	n.d.	n.d.	bulk	mRNA	microarray	PROS1
[14]	2015	resection	38	primary vs. recurrent	bulk	mRNA	WES	n.d.
[89]	2015	resection	96	n.d.	bulk	mRNA	microarray	CD151α3β1
[90]	2014	cell lines	n.d.	n.d.	bulk	mRNA	microarray	Dock7
[22]	2014	xenograft	n.d.	core vs. periphery	bulk	mRNA	microarray	e.g., neurophysiological processes, cell cycle
[91]	2014	cell lines	n.d.	n.d.	bulk	mRNA	microarray	LIMK1/2
[35]	2014	resection	5	n.d.	single-cell	mRNA	RNASeq	n.d.
[92]	2014	cell lines	n.d.	n.d.	bulk	mRNA	microarray	JAK2/STAT3
[93]	2013	resection	111	n.d.	bulk	mRNA	microarray	NFAT1
[94]	2013	resection	19	core vs. periphery	bulk	mRNA	microarray	NFκB
[95]	2012	cell lines	n.d.	n.d.	bulk	mRNA	microarray	FABP7
[96]	2012	cell lines	n.d.	core vs. periphery	bulk	mRNA	microarray	Galectin-1
[97]	2009	cell lines	n.d.	n.d.	bulk	mRNA	microarray	tGLI1
[98]	2008	cell lines	n.d.	n.d.	bulk	mRNA	microarray	PKCι, RhoB
[99]	2008	cell lines	10	core vs. periphery	bulk	mRNA	microarray	CTGF
[100]	2004	cell lines	n.d.	n.d.	bulk	mRNA	microarray	N-Cadherin
[101]	2003	cell lines	n.d.	n.d.	bulk	mRNA	microarray	IGFBP2

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
