# Peer review of "Tumor Cell Invasion in Glioblastoma"

_ijms, 2020, doi:10.3390/ijms21061932_

Round 1
Reviewer 1 Report
The review was well written and appropriate for the journal. The article covered a wide range of related to Glioblastoma Tumor cell invasion and various factor that influences the tumor microenvironment. I recommend the review for the publication provided authors to address the following comments.
1) The authors detailed the list of Single-cell and bulk cell analysis on GBM in the table1, they should include columns such as Omics signatures or driver genes identified in the listed studies.
2) The authors touched upon emerging technologies to address invasive GBM, notably the spatial Transcriptomics which can shed light on the immune cell invasion and relationship to the tumor microenvironment. The review should include more literature findings that came out recently using spatial transcriptomics study in different tissues and their potential directions to invasive-GBM.
Author Response
We thank the reviewers for the time and effort spent on our paper and for the constructive criticism. We addressed all comments and provide answers in the letter below and in the paper itself.
Reviewer 1
Comment 1:
“The authors detailed the list of Single-cell and bulk cell analysis on GBM in the table 1, they should include columns such as Omics signatures or driver genes identified in the listed studies.”
We thank the reviewer for this helpful suggestion. Accordingly, we added a new column to Table 1, listing genes or pathways associated with invasion identified in the respective publication (lane 135 ff).
Comment 2:
“The authors touched upon emerging technologies to address invasive GBM, notably the spatial transcriptomics which can shed light on the immune cell invasion and relationship to the tumor microenvironment. The review should include more literature findings that came out recently using spatial transcriptomics study in different tissues and their potential directions to invasive-GBM.”
The reviewer notes that emerging technologies, such as spatial transcriptomics, are only marginally touched. Spatial transcriptomic is a quite new and, at the same time, powerful technology. After the publication of Patrik L Ståhl et al. in Science 2016, 14 additional papers have been published on spatial transcriptomics of which 5 are more strictly related to the field of immunology and cancer research. Also, spatial transcriptomics has been applied on very limited numbers of tissues and cancers. We therefore have extended the literature on spatial transcriptomics and its application in other cancers. Studies about spatial transcriptomics and the inflammatory microenvironment have been also mentioned, as well as some possible applications in GBM (lane 241 ff).
Reviewer 2 Report
The authors summarized the progress on research of invasion in Glioblastoma including heterogeneity within glioblastoma, cell analysis, invasion mechanisms of invasion, microenvironment, new models and new technologies. The following are some concerns:
- Part 2 "Conserved genetic programs to coordinate invasion in glioblastoma". The authors commented that several genes common in normal as well as in in GBM progenitor cells have been described that shape cells to make them fit for invasion. The authors should combine these genes with more detailed information and describe how these genes work to help invasion. So the readers will have a clear picture.
- Part 3 "Heterogeneity and plasticity within glioblastoma". The authors should list the transcription factors which were identified to regulate the EMT, and then MET; Meanwhile, the author should describe how the heterogeneity drive GBM pathogenesis. So far, it is kind of confused.
- Part 4 "Bulk vs. single cell analysis". The authors listed the reports of the approaches for both bulk and single cell analysis which may related with invasion, genetic programmes and microenvironment. But, the authors didn't summarize the mechanisms which were elucidated by these publications. The authors need to clarify this part.
- Part 5 "Mechanisms of 118 invasion that relate to GBM cells". The authors listed some transcription factors like STAT3, NF-kB et al which may cooridinate the invasive programs. As this review is for invasion of GBM cells, please clarify is there any correlations between these invasive programmes and clinical progress?
- Part 8 "Emerging novel technologies and approaches for studying invasive GBM". In this part, the authors listed some new advances including spatial transcriptomics technology, scRNA-Seq, in silico-models, MRI et al., which were reported to help identify the behaviour of tumor invasiveness. The authors really need to emphasize how these new technologies could help to improve the diagnosis, prognosis predict and therapy. More detailed information are needed in this part.
- Part 9 "Overcoming treatment resistance by targeting invasion in GBM". The authors need to clarify how these signal pathways affect treatment resistance with more detailed information, instead of listing "reviewed in (164, 107 and 165)".
Author Response
We thank the reviewers for the time and effort spent on our paper and for the constructive criticism. We addressed all comments and provide answers in the letter below and in the paper itself.
Reviewer 2
Comment 1:
"The authors commented that several genes common in normal as well as in in GBM progenitor cells have been described that shape cells to make them fit for invasion. The authors should combine these genes with more detailed information and describe how these genes work to help invasion. So the readers will have a clear picture.
We agree with the reviewer and added functional information for the mentioned genes (line 69ff.)
Comment 2:
"The authors should list the transcription factors which were identified to regulate the EMT, and then MET; Meanwhile, the author should describe how the heterogeneity drive GBM pathogenesis. So far, it is kind of confused.”
We rearranged chapter 3 "Heterogeneity and plasticity within glioblastoma" (line 79 ff.) for less confusion. We also added a list of transcription factors for which a role in GBM EMT has been described (line 97 ff.) and how subtype plasticity is linked to GBM pathogenesis (line 106 ff.). A true MET doesn´t take place in GBM, but subtype switching back from a mesenchymal to e.g. a proneural phenotype has been observed, (PMID:28697342) and might well represent a mode of adaptation to conditions found in the new location of a disseminated glioma cell.
Comment 3:
"The authors listed the reports of the approaches for both bulk and single cell analysis which may related with invasion, genetic programmes and microenvironment. But, the authors didn't summarize the mechanisms which were elucidated by these publications. The authors need to clarify this part.”
This suggestion was very helpful and also made by reviewer 1. We added a new column to Table 1 (see above).
Comment 4:
"The authors listed some transcription factors like STAT3, NF-kB et al which may coordinate the invasive programs. As this review is for invasion of GBM cells, please clarify is there any correlations between these invasive programmes and clinical progress?
We feel unsure about the exact meaning of this comment, since „clinical progress“ in this context is ambigous to understand. However, since the reviewer refers to the mentioned transcription factors and invasive programs regulated by them, we deleted the respective short paragraph. It recapitulated factors and programs of EMT, which are now described in chapter 3 "Heterogeneity and plasticity within glioblastoma".
Comment 5:
“In this part, the authors listed some new advances including spatial transcriptomics technology, scRNA-Seq, in silico-models, MRI et al., which were reported to help identify the behaviour of tumor invasiveness. The authors really need to emphasize how these new technologies could help to improve the diagnosis, prognosis predict and therapy. More detailed information are needed in this part.”
We thank the reviewer for this comment. The reviewer has underlined the lack of more information for the mentioned techniques. We are aware that some arguments have been touched on a small extent with the intent to give a broader and general overview. As already said in our answer to reviewer 1, we have added additional literature and hints how these technologies could be applied to GBM to support diagnosis, prognosis and therapy.
Comment 6:
"The authors need to clarify how these signal pathways affect treatment resistance with more detailed information, instead of listing "reviewed in (164, 107 and 165)".
We agree with Reviewer 2 and detailed how different invasion-associated pathways relate to treatment resistance (line 293 ff.). However, since there is copious literature to the subject and discussion of all signalling pathways would exceed the scope of this review, we elaborate on two examples (p53 and integrins) only. Recent excellent reviews are available, which are still referred to.
Round 2
Reviewer 1 Report
The current revised version sufficiently addresses previous comments, added more literature information in the review. I recommend the current version for the publication.
Reviewer 2 Report
The authors made significant revisions. I don't have any comments.